# Experimental Reflection Evaluation for Attitude Monitoring of Space Orbiting Systems with NRL Arch Method

**Andrea Delfini [1,\*], Roberto Pastore [2], Fabrizio Piergentili [1], Fabio Santoni [2] and Mario Marchetti [2]**

1 Department of Mechanical and Aerospace Engineering, Sapienza Università di Roma, Via Eudossiana 18, 00184 Rome, Italy; fabrizio.piergentili@uniroma1.it
2 Department of Astronautics, Electric and Energy Engineering, Sapienza Università di Roma, Via Eudossiana 18, 00184 Rome, Italy; roberto.pastore@uniroma1.it (R.P.); fabio.santoni@uniroma1.it (F.S.); mario.marchetti@uniroma1.it (M.M.)
\* Correspondence: andrea.delfini@uniroma1.it

**Abstract:** The increasing number of satellites orbiting around Earth has led to an uncontrolled increase in objects within the orbital environment. Since the beginning of the space age on 4 October 1957 (launch of Sputnik I), there have been more than 4900 space launches, leading to over 18,000 satellites and ground-trackable objects currently orbiting the Earth. For each satellite launched, several other objects are also sent into orbit, including rocket upper stages, instrument covers, and so on. Having a reliable system for tracking objects and satellites and monitoring their attitude is at present a mandatory challenge in order to prevent dangerous collisions and an increase in space debris. In this paper, the evaluation of the reflection coefficient of different shaped objects has been carried out by means of the bi-static reflection method, also known as NRL arch measurement, in order to evaluate their visibility and attitude in a wide range of frequencies (12–18 GHz). The test campaign aims to correlate the experimental measures with the hypothetical reflection properties of orbiting systems.

**Keywords:** attitude monitoring; NRL arch; reflection coefficient; EM characterization

## 1. Introduction

The determination of the attitude (i.e., the orientation with respect to a given frame of reference) of satellites and orbiting objects is one of the most important tasks for present-day space safety [1–3]. The ever-increasing quantity of space debris, therefore, imposes an increasingly pressing need to evaluate the trajectory and effects of these potentially dangerous objects. Regarding this critical issue, it is essential to remember that in 2014 the European Commission, conscious of the present urgency, undertook the development of a European network of sensors for the surveillance and tracking of orbiting objects and initiated a specific SST (Space Surveillance and Tracking) support framework program. Italy, Germany, the U.K., France, and Spain joined the program and constituted, with SatCent, the front desk for SST services, the EUSST Consortium.

In this frame, with several thousand objects orbiting around Earth, having a tool similar to a radar system that could help to track objects and determine their attitude and re-entry trajectory is therefore of primary importance. In order to envisage the trajectory of these objects, one has to know their attitude to evaluate the effects of the atmospheric drag on the trajectory itself, also associating radar measurements with other tracking systems such as the optical system, for example, LED (Light Emission Diodes) [4–7], or light-curve acquisition systems [8–10] or magnetometer data [11]. Knowing the attitude using radar systems, therefore, becomes one of the fundamental tasks for the detection of space debris, as demonstrated by the recent case of the Chinese Space Station Tiangong-1 [12].

To this aim, the bi-static reflection method, also known as NRL arc measurement, was used in order to determine the reflectivity of several objects that can be assimilated to space debris or to an orbiting satellite, relating the reflectivity to the object's attitude. NRL arch is the industry standard for testing the reflectivity of materials. Originally designed at the Naval Research Laboratory (NRL), the NRL arch enables fast, repeatable, and non-destructive testing over a wide range of frequencies [13–15]. The experimental reflection results have been related to real radar tracking conditions. The reference radar for tracking orbiting satellites has a frequency of 400 MHz.

In the NRL system, two antennas are used, one for transmitting and the other for receiving signals, from a vector network analyzer (VNA), and microwave reflectivity can be measured at different angles of incidence, simulating different attitude positions. Several objects were built in aluminum, in scale with respect to a real satellite, in order to evaluate the reflection in different conditions of attitude and over a wide range of frequencies, between 12 and 18 GHz, relating the results to the real frequencies and dimensions of use.

## 2. Materials and Methods

The objects under investigation are aluminum-covered models created in order to evaluate their reflection properties in different attitude positions and over a wide range of frequencies. In this case, the attitude is the orientation of the object with respect to its hypothetical orbit. The test campaign, thus, aims to correlate the real experimental measures with the reflection properties of hypothetical orbiting objects.

This aim is carried out considering a reference radar working at 400 MHz, with a wavelength of 75 cm. The experimental frequency range for measurements was set to 12–18 GHz with wavelengths of a maximum of 2.5 cm and a minimum of 1.7 cm.

The relation between the real experimental data and the predicted reflection properties is thus given by a simple proportion between the samples' size, the experimental wavelength, and the reference radar wavelength: a given experimental frequency (12 GHz as an example) corresponds to a given wavelength (2.5 cm), with a precise ratio to the sample size. The same ratio is considered when a 400 MHz, 75 cm wavelength is applied, finding the hypothetical real object size, which is different for every frequency in the 12–18 GHz span. In other words, as the frequency of the reference radar is 400 MHz, with a wavelength of 75 cm, the scaling process of the samples, as a first approximation (i.e., considering only geometry and shape, without considering the effects of the atmosphere such as reflection, refraction, diffraction and interference that are anyway present in a ground tracking), allows to correlate the reflection properties of tested objects, shown in Table 1, to real size objects. In Figure 1, the experimental setup is shown.

The choice of an experimental campaign based on a frequency span measure method and not on a single frequency can be explained with the flexibility of such a methodology, which allows a wider relation between samples and hypothetical real objects. Moreover, the size of the samples was chosen considering the incident wavelength: in real tracking, at 400 MHz, the wavelength is approximately of the same order of magnitude of the satellite; thus, the same relation was considered for the samples. The shape of the samples was chosen considering the most common shapes of satellites. In future works, a complete satellite model will be manufactured, with more complex geometries, such as parabolic antennas.

The NRL arch method was chosen for the experimental campaign for its capability to perform free space measures: the incident signal wavelength is extremely lower than the distance between antennas and the target, simulating the real signal transmission in the best possible way.



**Table 1.** Pictures and characteristics of the samples under investigation, having a different geometry in order to evaluate different hypothetical orbiting objects. Wavelengths of a maximum of 2.5 cm and a minimum of 1.7 cm were considered for the measurements and scaling process.

| Specimen | Dimensions |
| :---: | :---: |
| 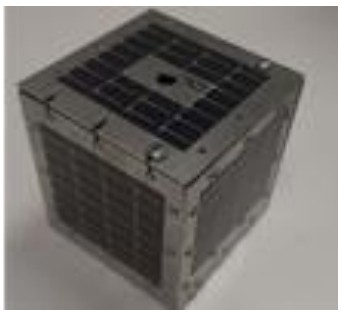 | Cubesat Characteristics<br>Dimensions of 0.1 m per side,<br>attributable respectively to a satellite of 3 m per side at 12 GHz and to one of 4.4 m at 18 GHz. |
| 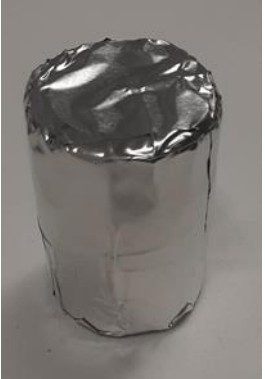 | Cylinder Characteristics<br>Dimensions:<br>Height: 0.075 m, Diameter: 0.065 m<br>attributable respectively to a satellite of 2.25 m in height at 12 GHz and to one of 3.3 m at 18 GHz. |
| 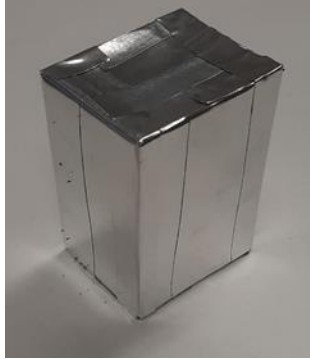 | Parallelepiped Characteristics<br>Dimensions:<br>Height: 0.08 m, Width: 0.06 m, Thickness: 0.04 m<br>attributable respectively to a satellite 2.4 m high at 12 GHz and one of 3.5 high at 18 GHz. |
| 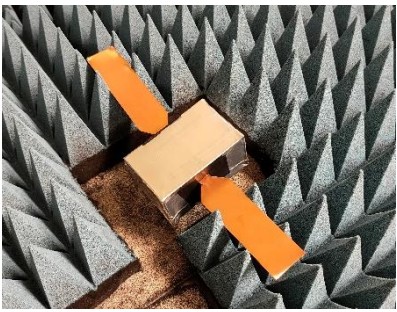 | Parallelepiped Characteristics<br>plus<br>Appendices size:<br>$0.12 \times 0.02$ each, the whole object corresponding to satellites with a body 2.4 m high at 12 GHz and a body 3.5 m high at 18 GHz, with appendices spanning 3.6 m and 5.3 m, respectively. |

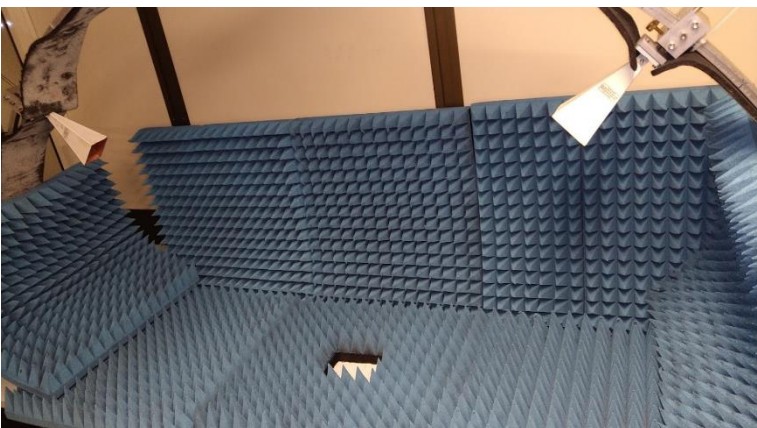

**Figure 1.** NRL ARCH setup, with horn antennas and anechoic panels.

Reflectivity is defined as the reduction in reflected power caused by the introduction of a material [16–18]. This power reduction is compared to a "perfect" reflection, which comes very close to the reflection of a flat metal plate. The antennas can be positioned anywhere on the arc to allow performance measurements with angles of incidence not normal to the sample.

The vector network analyzer is used to provide both stimulus and measurement [19]. In the present case, considering that the aim is to evaluate the reflection of metal objects in free space, the calibration is performed by measuring the resulting power reflecting off a metal plate over a wide frequency range and then over the same frequency range on a radar absorbing material (ECCOSORB HPY-60, with a reflectivity of −50 dB). The latter measure will be the "zero" or 0 dB level (i.e., the reference level), considering the net of errors in the NRL measurement setup, in order to simulate a radio wave that is lost in space without encountering any reflecting body [20]. The material under test is then placed on the radar absorbing material (RAM) plate, and the reflected signal is measured in dB. The time-domain gate and anechoic panels are used to eliminate antenna cross-talk and clear the error otherwise introduced by room reflections as well as noise. Using this configuration, it is possible to characterize the properties of systems in different directions. The measurement system is based on Agilent 8571E software (material measurement) and the Agilent PNA-L N5230C vector analyzer. The antennas are Q-par Angus Ltd. and are active in the 12–18 GHz range. The sample rating was set at 512 points, with a power of −15 dBm and a 1 kHz bandwidth, and the TE polarization of antennas was considered.

The measurements' reliability lies within the 2 dB range with respect to the reflection properties declared in the ECCOSORB datasheet. Measurements of the fabricated samples are then performed. The antennas are placed at a 45° angle with respect to the sample (Figure 1). Every measure has been smoothed with a polynomial (for CubeSat and cylinder) and a mobile average (for parallelepiped and parallelepiped with appendices) trend line for a better comprehension of the VNA response.

## 3. Results and Discussion

### 3.1. Cubesat

As reported in Table 1, the sample considered is 0.1 m per side, with a minimum wavelength of 0.025 m and a maximum of 0.017 m that refers to a satellite of 3 m per side at 12 GHz and to one of 4.4 m at 18 GHz. The positioning of the CubeSat and the measures are shown, respectively, in Figures 2 and 3. Figure 2 shows only one antenna, the sender, in order to indicate the direction of the signal; the receiving antenna is not shown in the sketch (the whole system can be visible in Figure 1).

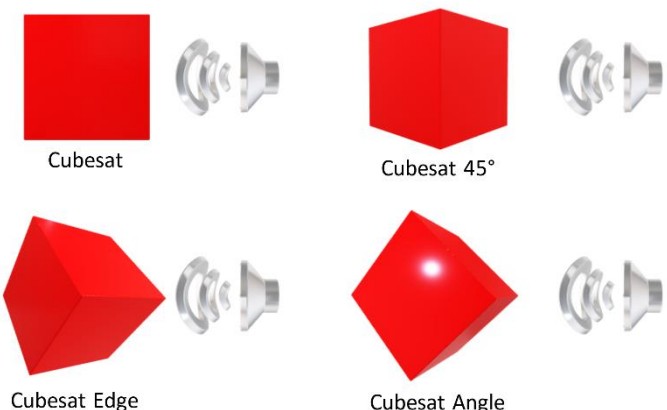

**Figure 2.** Cubesat positioning in respect to wave incidence direction.

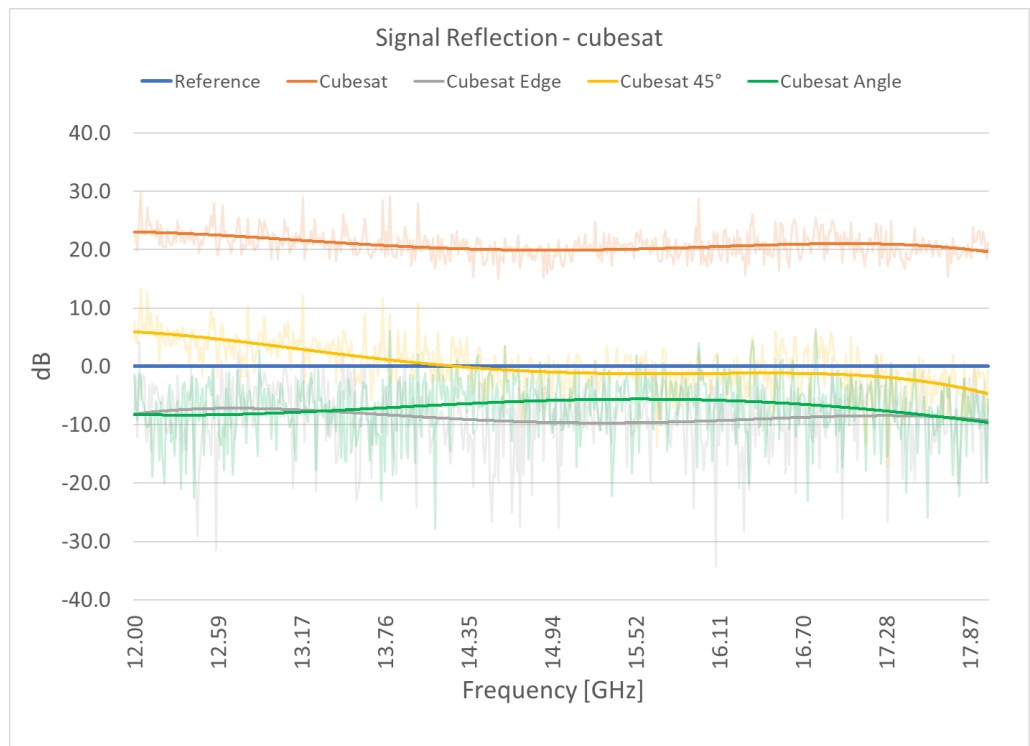

**Figure 3.** Cubesat signal reflection: TE polarization reflection vs. frequency.

The configuration of the greatest reflection is the "flat" one. For the other measurements, it can be seen that the CubeSat 45° configuration reflects between 12 and 13.8 GHz.

The results lead to the consideration that scaled objects in the flat position and within the dimensions range aforementioned are visible with a 400 MHz radar system. Moreover, that means that satellites between 3 and 3.8 m can be seen when they are in the CubeSat 45° attitude configuration. The presence of negative curves (meaning more absorption) testifies that in those configurations, the signal is diverted toward the absorbing plane and therefore does not reach the receiving antenna.

### 3.2. Cylinder

The sample has the following parameters. Height: 0.075 m, Diameter: 0.065 m, minimum frequency 0.025 m, and maximum 0.017 m that can be referred to a satellite of 2.25 m in height at 12 GHz and to one of 3.3 m at 18 GHz. Three configurations were considered. The first, cylinder in a vertical position; the second, cylinder in a horizontal position and rotating around the vertical axis; the third, cylinder in precession, according to

a cone of 45° with respect to the vertical axis. In this configuration, the 0° angle is the one in which the satellite is aligned with the incident wave. The positions 45°, 90°, 315° have been considered. In Figure 4, the positioning of the samples is shown, while in Figure 5, the reflection plots are given.

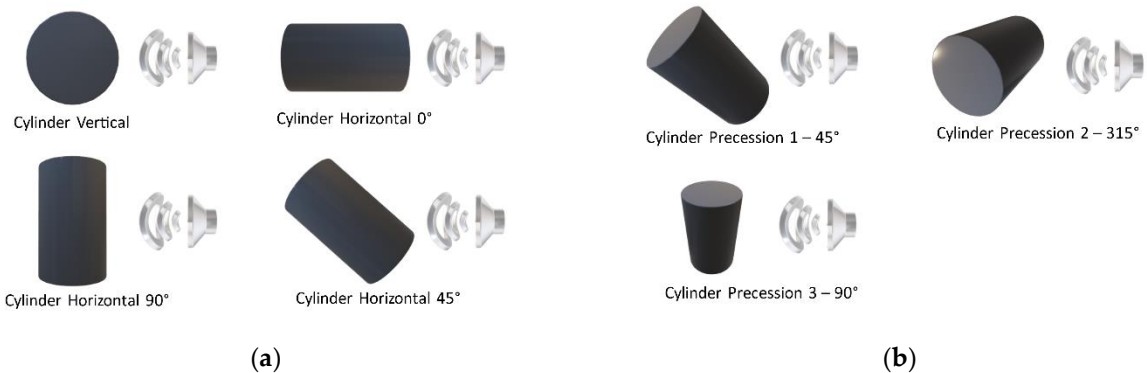

**Figure 4.** Cylinder positioning in respect to wave incidence direction. (**a**) Horizontal and vertical positions; (**b**) precession positions.

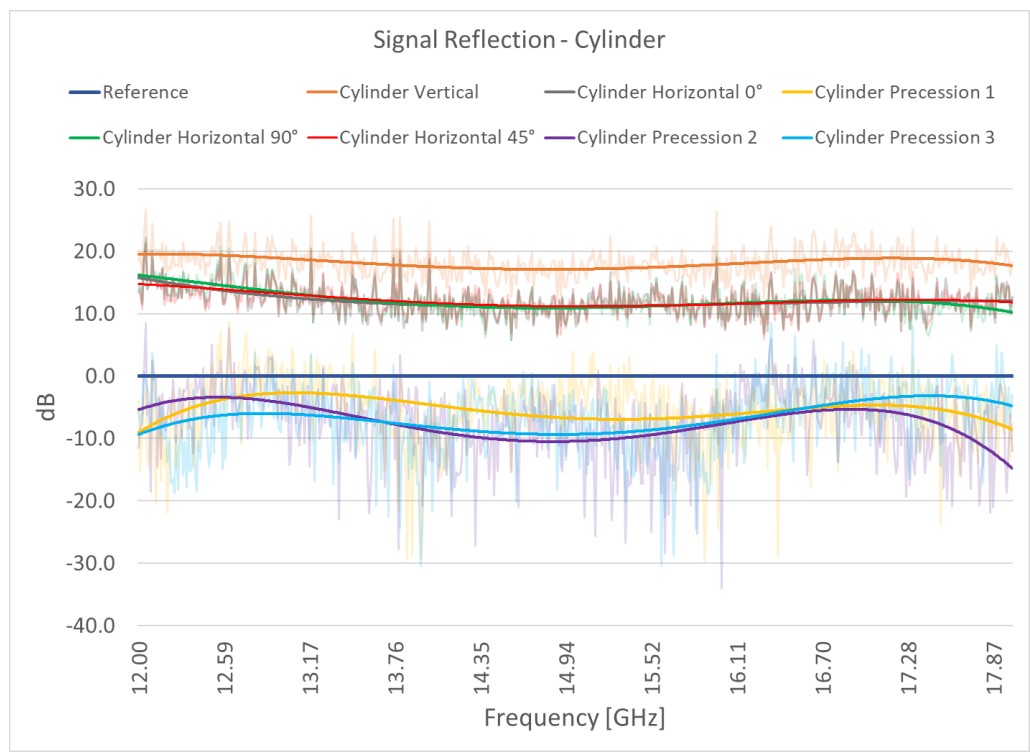

**Figure 5.** Cylinder signal reflection: TE polarization reflection vs. frequency.

In this case, too, the greatest reflection occurs in the vertical configuration, followed by the three horizontal configurations. The object is clearly visible throughout the frequency range. The precession configurations, on the other hand, are, at these frequencies, completely invisible to the radar, as their reflection does not occur in the direction of the receiving antenna.

Regarding the scaled objects, the result also leads to the consideration that satellites with the dimensions range aforementioned are visible with a 400 MHz radar system when in a vertical or horizontal attitude, while the precession configurations are completely invisible to the radar.

*3.3. Parallelepiped*

The sample has the following parameters. Sample size: $0.08 \times 0.06 \times 0.04$ m, minimum frequency 0.025 m, and maximum 0.017 m attributable to a satellite 2.4 m high at 12 GHz and one of 3.5 at 18 GHz. Three configurations were considered. The first, satellite in a vertical position; the second, satellite in a horizontal position and rotating around the vertical axis; the third, satellite in a precession, according to a 45° cone with respect to the vertical axis. In this configuration, the 0° angle is the one in which the satellite is perpendicular to the incident wave. The positions 45°, 225°, 270°, 180°, and 225° were considered with an anticlockwise rotation of 45° around the longitudinal satellite axis. In Figure 6, the positioning of the sample is shown; in Figure 7, the reflectivity plots are depicted.

In this case, the maximum reflection occurs in the configurations in horizontal rotation, which exposes a greater surface to the signal; the maximum reflection occurs for the 90° configuration, followed by the 45° and the 0° configuration. The vertical configuration also reflects the signal well. It can be seen that the shape of the curves is absolutely unchanged, with an almost constant frequency response.

In the configurations of the samples in precession, the only one that shows a non-reflective behavior in all frequencies is 1, with 225° of rotation around the precession axis, in which the signal is reflected in all directions except toward the receiving antenna. In the others, it can be seen how configuration 5, at 135° of rotation around the precession axis, is the one that manages to reflect the signal in all frequencies, with peaks between 5 and 6 dB, at regular intervals, at frequencies 13.8, 15.5 and 17 GHz. At 15 GHz, there is the minimum, where there is no reflection. It will thus be possible to understand when the satellite will be in this configuration by associating an optical detection system. Configuration 3, at 270° of rotation around the precession axis, also has frequencies in which it is possible to identify its position, as well as 2 and 0 (180° and 45°). Respectively, this occurs at frequencies between 12.4 and 14 GHz (3), between 15.5 and 16.3, and at 18 GHz (2), at 12.9, and between 14.0 and 14.4 GHz (0). Configuration 4, 225° of rotation around the precession axis with 45° rotation around the longitudinal axis of the satellite, has suitable reflection only at 12.4 GHz.

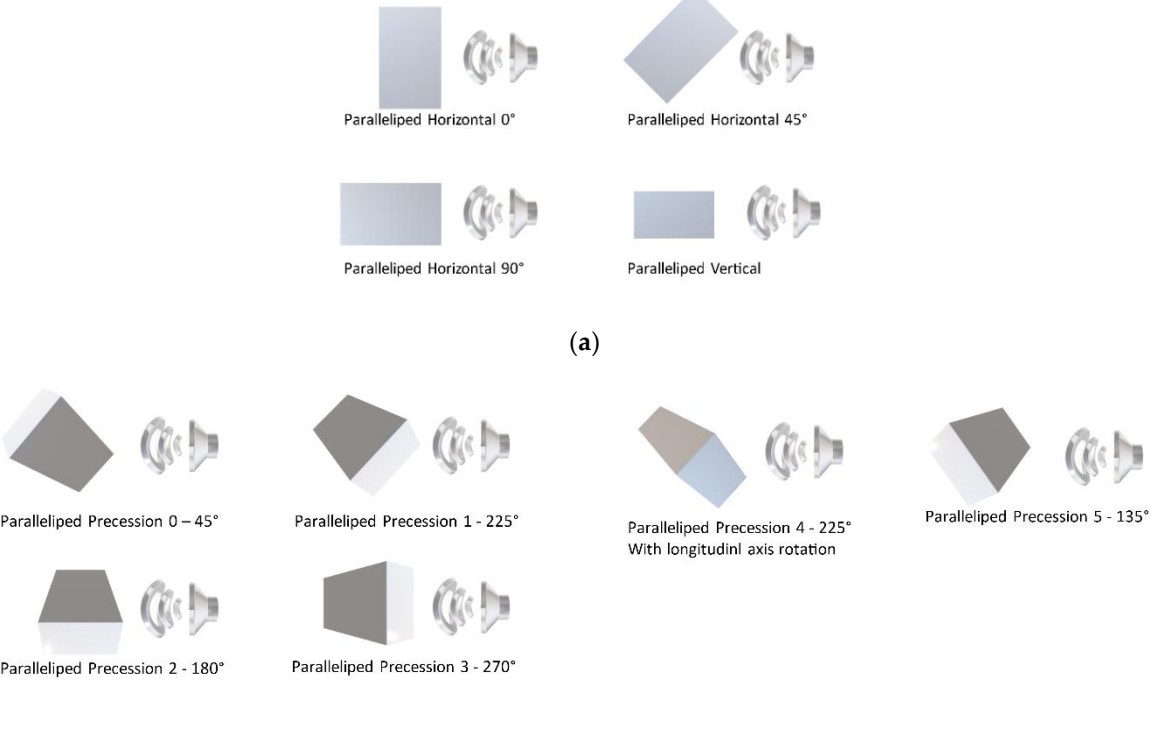

**Figure 6.** Parallelepiped positioning in respect to wave incidence direction. (**a**) Horizontal and vertical positions; (**b**) precession positions.

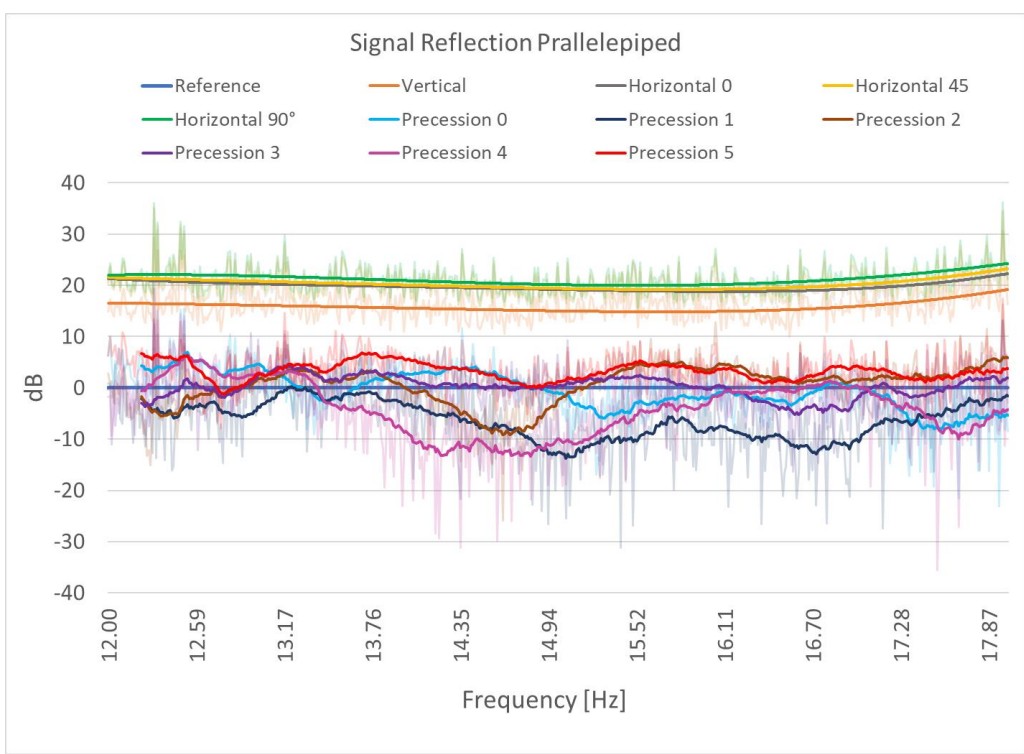

**Figure 7.** Parallelepiped signal reflection: TE polarization reflection vs. frequency.

The experimental results show that scaled objects in the 90°, 45°, and 0° attitude positions, with the dimensions, range aforementioned, are visible with a 400 MHz radar system.

Considering the precession positions where the objects better reflect the incident waves, it can be assumed that the corresponding scaled object of 3.5 m will be visible only in some positions when a 400 MHz radar signal is tracking it: it will be visible for all the horizontal and vertical positions and for Precession 2, Precession 3 and Precession 5 positions. Considering the differences in the reflected values, the different positions can be clearly found.

Considering the 18 GHz frequency (3.5 m scaled object), the reflection signal related to the relative attitude positions can be highlighted in Figure 8, where both planar positions and precession positions are shown.

The great visibility of the object can be noted when it is placed in horizontal positions and rotation, while, when in precession, it appears clear that the visibility strongly depends on the relative position. Moreover, the reflection trend for the horizontal position leads to the consideration that the larger is the width of the body exposed to the EM field, the higher is the reflection and thus the visibility. On the other hand, as said, considering the object in precession, the relative position of the body plays a more important role: the orientation due to the rotation is responsible for the highest visibility at 270°, as the path of the reflected wave is directed toward the receiving antenna, and it is not scattered away as in the 225° position.

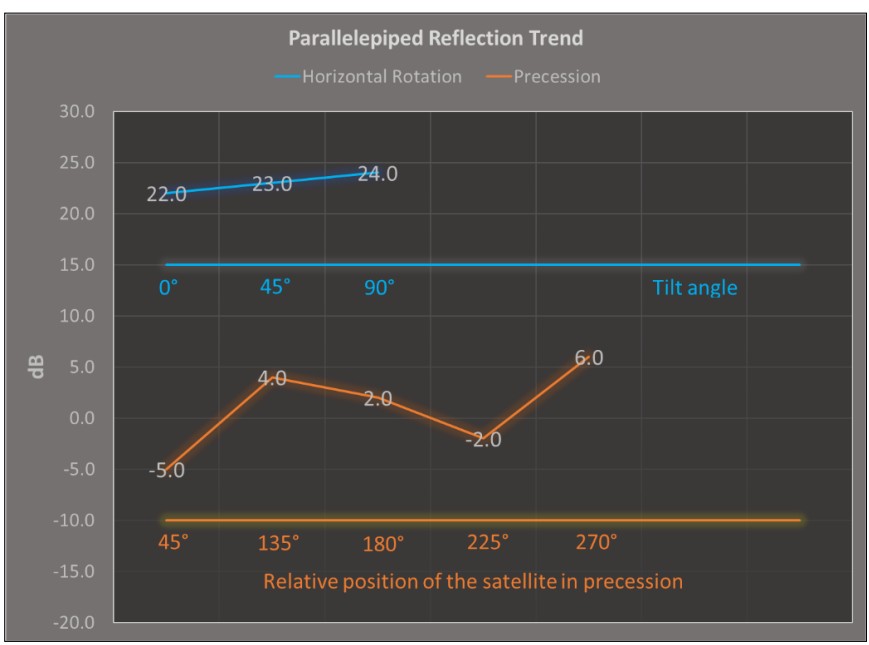

**Figure 8.** Parallelepiped reflection trend.

### 3.4. Prallelepiped with Appendices

Three configurations were considered. The first, object in a vertical position; the second, object in a horizontal position and rotating around the vertical axis; the third, sample in a precession, according to a 45° cone with respect to the vertical axis. In this configuration, an anticlockwise rotation was considered, with 0°, 45°, 90°, 135°, 270°, and 315° positions. The positioning of the sample and the measures graphs are shown in Figures 9 and 10, respectively.

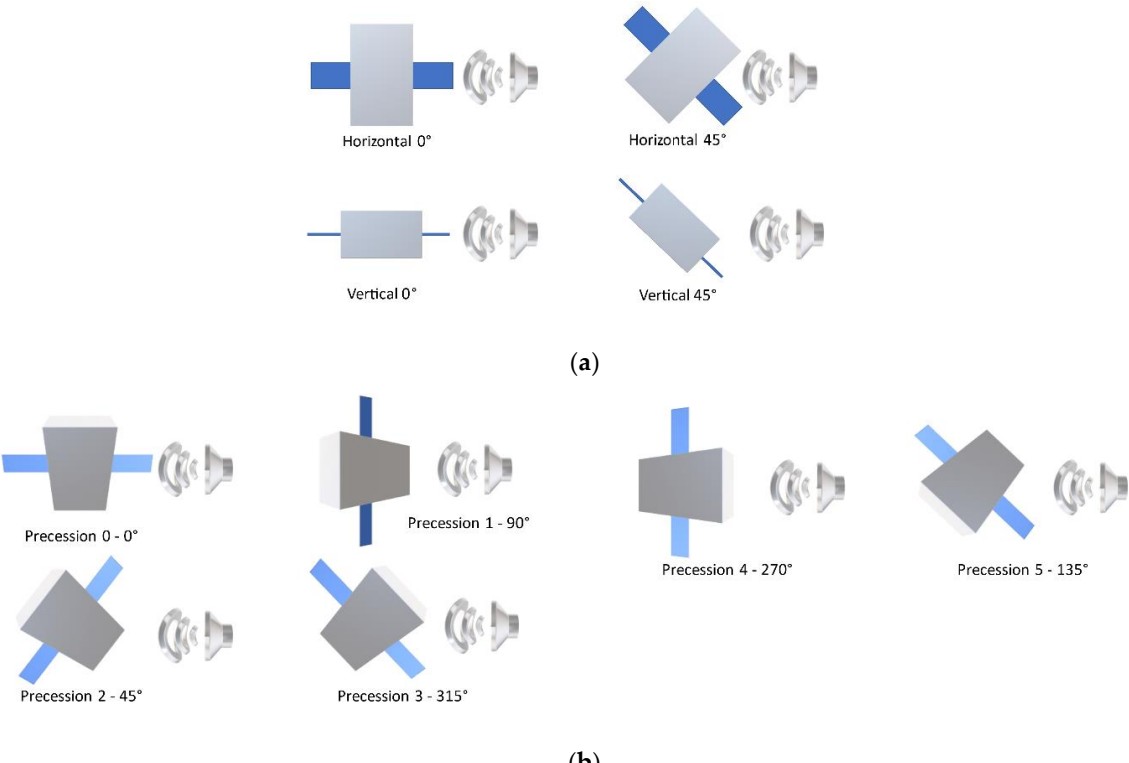

**Figure 9.** Parallelepiped with appendices positioning in respect to the wave incidence direction. (**a**) Horizontal and vertical positions; (**b**) precession positions.

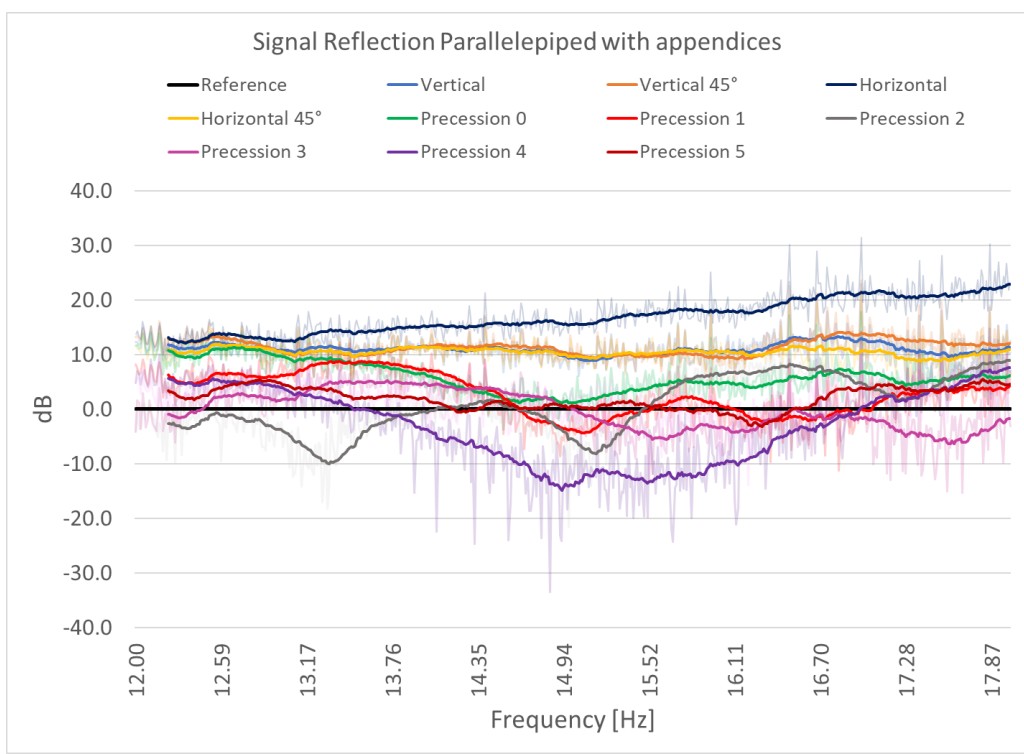

**Figure 10.** Parallelepiped with appendices signal reflection: TE polarization reflection vs. frequency.

The case with the highest reflection is the horizontal position, with an increasing trend between 13 and 23 dB of reflection at the extreme frequencies. Vertical, vertical at 45° of rotation on the longitudinal axis, and horizontal at 45° of rotation on the longitudinal axis measures have a more flattened trend starting from 13 dB at 12 GHz and ending at 10 dB at 18 GHz. It is also important to note that several spikes on the plots are visible for all the positions at the same frequencies (16.5 GHz, 16.7 GHz, 17 GHz, 17.4 GHz, 17.8 GHz).

All the precession positions are visible at 18 GHz, except position Precession 3. From 15.5 GHz to 18 GHz, satellites in Precession 0 and Precession 2 are always visible, while objects in the Precession 1 position are visible between 17.3 and 18 GHz only.

At lower frequencies, from 12 to 13.8 GHz, satellites in the Precession 2 position are not visible, while Precession 0 and 1 present the highest values of reflection.

The test campaign leads to the consideration that scaled objects, in horizontal, vertical, vertical at 45° of rotation on the longitudinal axis, and horizontal at 45° of rotation on the longitudinal axis positions, with the dimensions range aforementioned, are visible with a 400 MHz radar system.

That means that, for instance, an object of 3.46 m at 17.8 GHz, with relative appendices, is clearly visible. For precession positions, an object of 3.5 m length is always visible, except when in the Precession 3 position and its attitude is determined by its reflection value. Objects between 3 and 3.5 m are visible when in Precession 0 and Precession 2 positions, while when in the Precession 1 position, only objects between 3.4 and 3.5 m are visible. Objects from 2.4 to 2.7 m in the Precession 2 position are not visible, while Precession 0 and 1 present the highest reflection.

As a concluding remark, in view of the differences in the above-mentioned reflected values, the different attitude positions can be clearly found for the objects under consideration when they are tracked with a 400 MHz radar signal.

Regarding the 18 GHz frequency (3.5 m scaled object), the reflection signal related to the relative attitude positions is highlighted in Figure 11, where both planar and precession positions are shown.

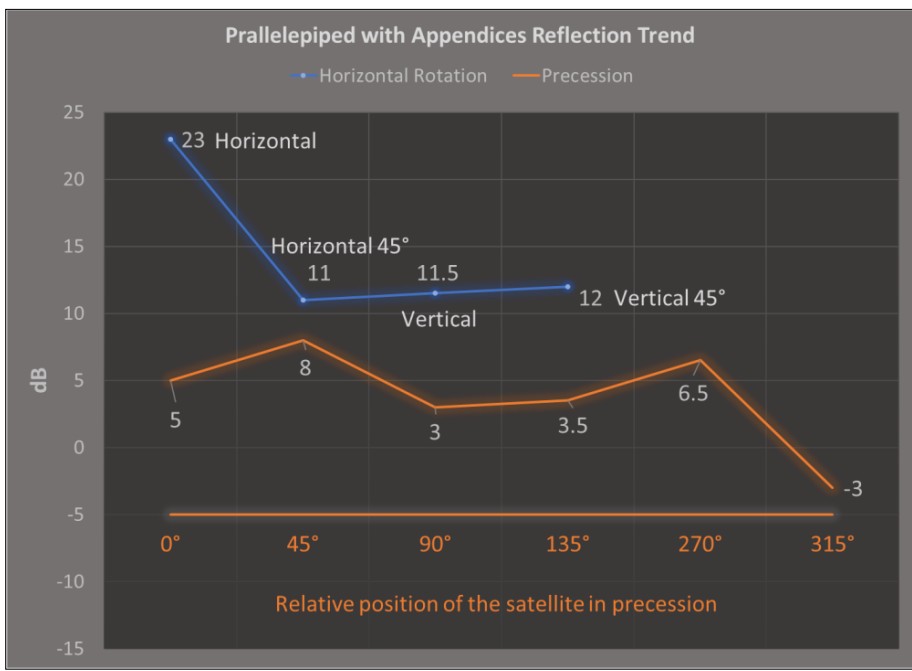

**Figure 11.** Parallelepiped with appendices reflection trend.

The great visibility of the object can be noted when it is placed in horizontal positions and rotation; instead, when in precession, it appears clear that the visibility depends on the relative position but, thanks to the presence of the appendices, only the 315°— Precession 3 position is not visible. The reflection trend for the in-plane positions leads to the consideration that, because the larger is the width of the body exposed to the EM field, the higher is the reflection and thus the visibility, the presence of the appendices enhances the reflection properties of the body in horizontal 0° position. In the horizontal 45° position, the reflected wave is not affected by the presence of the appendices, so that the vertical positions present almost the same value making it difficult to identify the relative position.

On the other hand, considering the object in precession, the body relative position plays the most important role: the orientation due to the rotation is responsible for the highest visibility at 45°. Differently from the previous case, the 270° position, although with the second higher reflection value, presents a slightly lower value, probably due to the positions of the appendices in respect to the incident wave.

## 4. Conclusions

The test campaign has brought very promising results. The behavior of the samples, tested in different shapes and configurations, showed that it is possible to determine the attitude in orbit of an object based on its reflection from a radar signal. In fact, the different configurations tested show how a different setup produces a single response, which can be associated with a different position. By associating this system with an optical system, for example, LEDs or light-curve acquisition systems, or magnetometer data to determine the attitude of a satellite, it will be possible to determine its exact position. Scaling the model makes it possible to carry out evaluations even for large satellites and space debris, which are usually identified by radar systems with much lower frequencies than those of the experimental system considered.

These results, therefore, have a double value: they allow us to identify small satellites in a high-frequency range (12–18 GHz) as well as to have a prediction of visibility of large orbiting systems at the frequencies of the actual present radar systems, based on the scale considered. For example, it will be possible to determine the attitude of a satellite of a certain size by considering the working frequency of the radar and its wavelength and a scale model that is subjected to a measurement frequency at the same scale.

**Author Contributions:** Conceptualization, A.D., F.P.; methodology, A.D. and R.P.; software, A.D. and R.P.; validation, F.S., F.P. and M.M.; formal analysis, A.D. and R.P.; investigation, A.D., F.P. and R.P.; resources, F.P.; data curation, A.D. and R.P.; writing—original draft preparation, A.D. and R.P.; writing—review and editing, A.D., F.P., F.S., M.M. and R.P.; visualization, A.D.; supervision, F.P. and F.S.; project administration, F.P.; funding acquisition, F.P. All authors have read and agreed to the published version of the manuscript.

**Funding:** This research was funded by the Italian Space Agency through the grant agreement n. 2020-6-HH.0 (Detriti Spaziali—Supporto alle attività IADC e SST 2019–2021).

**Conflicts of Interest:** The authors declare no conflict of interest.

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
