# Peer review of "Experimental Reflection Evaluation for Attitude Monitoring of Space Orbiting Systems with NRL Arch Method"

_applsci, doi:10.3390/app11188632_

Round 1
Reviewer 1 Report
This is the referee report of the paper “Experimental Reflection Evaluation for Attitude
Monitoring of Space Orbiting Systems with NRL Arch Method“.
The authors Andrea Delfini et al. conducted reflection experiments in order to develop a potential tool for tracking objects and satellites orbiting around Earth. The motivation for such a study is the uncontrolled increase in the number of orbital objects in the last decades and the necessity to monitor these objects.
However, after giving the matter a great deal of thought, I can’t recommend the presented study for publication in the journal “Applied Sciences”. In my opinion, the presented study doesn’t provide any significant new contributions to the mentioned problem. The authors conducted experiments and try to analyse the experimental data, but I have several concerns about this project.
The motivation of the project is good. It makes absolutely sense to monitor objects orbiting the Earth. The authors also say several times they want to monitor the “attitude” of the satellite, however, they don’t describe what they mean with the term “attitude”. Do they want to measure the properties of the objects, like size, shape, or direction? This is unclear, from the beginning to the end.
It makes sense to start with this study in the laboratory and to illuminate several objects, but the connection between the experiments and the initial problem is absolutely unclear. Why do the authors choose the four shapes, and not other ones? Why do they take these sizes, and not smaller or larger ones? What is the kind of the illumination source? Why is the reference radar frequency 400 MHz, and the frequencies in the experiment are 12-18GHZ? Why a single frequency versus a range of frequency?
There is no physical argumentation or even a proof why we can simply scale up objects of laboratory size to the size of ~3m. Did the authors thought about distances? The distance between illumination source and the sample is in the order of one meter or less, the distances of objects in the earth orbit is hundreds of km. Why is it possible that we can still observe these objects? What about effects in the atmosphere? What does that mean for the illumination source and for the detector sensitivity? The only criterion for observability is a signal > 0 compared to the reference case, but what about noise?
The authors mention that the detector is located on an arch, however, the Figures 2,4,6,9 show the detector position just in the same direction as the sender. The real set up of the experiment is unclear. It is also unclear what the measured signal really is. What is the impact of some general properties on the signal, e.g. distance of the object, the material, other shapes. A spherical shape would be of interest too. What is a “relative image”?
Unfortunately, the paper is not well written. Several sentences are redundant or are given multiple times (e.g. “… the eccosorb plate is taken as a reference for the measurement, in order to simulate a radio wave that is lost in a vacuum without encountering any reflecting body”). The language has to be improved significantly, and the quality of the figures doesn’t fit to the requirements of a refereed journal.
In summary, there are a lot of open questions and issues. I am very sorry, but given the state of the paper I can’t recommend for publication and have to reject the paper.
Kind regards,
Anonymous referee
Reviewer 2 Report
153. Remove point.
156. Remove space.
159 - 163 and below. Possibly, not needed space between number and degree symbol.
238. remove comma.
The chosen methodology and the presented results look reliable and are important for improving the capabilities of satellite detection and their orbit determination tasks.
Reviewer 3 Report
The paper “Experimental Reflection Evaluation for Attitude Monitoring of Space Orbiting Systems with NRL Arch Method”, by Mario Marchetti, Andrea Delfini, Roberto Pastore, Fabio Santoni and Fabrizio Piergentili studies the problem of evaluating the visibility and attitude of objects in space, in a wide range of frequencies (12-18 GHz). A test campaign was made to correlate the experimental measures with hypothetical orbiting systems reflection properties.
The paper is interesting and gives new results in the problem of evaluating the visibility and attitude of objects in space, with a very practical approach. The quality of the figures is good.
Regarding technical aspects, the paper describes the problem and the solution proposed, but I still have some questions that may improve the paper. They are:
- I believe some more information about the effects of the angular rotation of the bodies in the evaluation of the attitude would be interesting;
- Tests were made using simple geometrical forms. It would be nice to know if the technique works for more complex geometries, in particular when there are convex and concave parts.
Both suggestions have the goal of showing the applicability of the method, which I believe is very important in a practical research like this one.
So, in general, I consider that this is a good paper and can be published after some adjustments.
Round 2
Reviewer 3 Report
I consider that all the questions I raised were adequately answered and the paper is now ready for publication from the technical point of view.